# Percutaneous Endoscopic Gastrostomy versus Nasogastric Tube Feeding: Oropharyngeal Dysphagia Increases Risk for Pneumonia Requiring Hospital Admission

**DOI:** 10.3390/nu11122969

**Published:** 2019-12-05

**Authors:** Wei-Kuo Chang, Hsin-Hung Huang, Hsuan-Hwai Lin, Chen-Liang Tsai

**Affiliations:** 1Division of Gastroenterology, Department of Internal Medicine, Tri-Service General Hospital, National Defense Medical Center, Taipei 114, Taiwan; xinhung@gmail.com (H.-H.H.); redstone120@yahoo.com.tw (H.-H.L.); 2Division of Pulmonary and Critical Care, Department of Internal Medicine, Tri-Service General Hospital, National Defense Medical Center, Taipei 114, Taiwan; doc10376@gmail.com

**Keywords:** oropharyngeal dysphagia, aspiration pneumonia, percutaneous endoscopic gastrostomy, nasogastric tube

## Abstract

Background: Aspiration pneumonia is the most common cause of death in patients with percutaneous endoscopic gastrostomy (PEG) and nasogastric tube (NGT) feeding. This study aimed to compare PEG versus NGT feeding regarding the risk of pneumonia, according to the severity of pooling secretions in the pharyngolaryngeal region. Methods: Patients were stratified by endoscopic observation of the pooling secretions in the pharyngolaryngeal region: control group (<25% pooling secretions filling the pyriform sinus), pharyngeal group (25–100% pooling secretions filling the pyriform sinus), and laryngeal group (pooling secretions entering the laryngeal vestibule). Demographic data, swallowing level scale score, and pneumonia requiring hospital admission were recorded. Results: Patients with NGT (*n* = 97) had a significantly higher incidence of pneumonia (episodes/person-years) than those patients with PEG (*n* = 130) in the pharyngeal group (3.6 ± 1.0 vs. 2.3 ± 2.1, *P* < 0.001) and the laryngeal group (3.8 ± 0.5 vs. 2.3 ± 2.2 vs, *P* < 0.001). The risk of pneumonia was significantly higher in patients with NGT than in patients with PEG (adjusted hazard ratio = 2.85, 95% CI: 1.46–4.98, *P* < 0.001). Cumulative proportion of pneumonia was significantly higher in patients with NGT than with PEG for patients when combining the two groups (pharyngeal + laryngeal groups) (*P* = 0.035). Conclusion: PEG is a better choice than NGT feeding due to the decrease in risk of pneumonia requiring hospital admission, particularly in patients with abnormal amounts of pooling secretions accumulation in the pyriform sinus or leak into the laryngeal vestibule.

## 1. Introduction

Percutaneous endoscopic gastrostomy (PEG) and nasogastric tube (NGT) feeding are widely used for delivering enteral nutrition to patients with dysphagia [1,2]. Current guidelines recommend PEG for long-term and NGT for short-term enteral nutrition [3,4,5]. However, patients are commonly placed on NGT for long-term enteral feeding in Asian countries [6,7,8].

Aspiration pneumonia is the most common cause of death in patients with PEG and NGT feeding [9,10]. Meta-analyses of pneumonia risk did not observe that patients with NGT were at higher risk than those with PEG, but these results may be due to high levels of statistical heterogeneity between studies [2,11,12]. Subgroup analysis is required to explore the variation across studies.

Oropharyngeal dysphagia has been identified as a serious risk factor for patients developing aspiration pneumonia [13,14]. Endoscopy allows for direct observation of food accumulation in the pyriform sinus and other risks such as food entering below the vocal cords, which is common with oropharyngeal dysphagia [15,16,17]. With upper gastrointestinal (UGI) endoscopy, clinicians can observe the amount and location of accumulated pooling secretions in the pharyngolaryngeal region [17,18,19,20].

The aim of this study was to compare the risk of developing pneumonia requiring hospital admission between individuals using PEG versus NGT for long-term feeding according to UGI endoscopic observation of the severity of pooling secretions in the pharyngolaryngeal region.

## 2. Materials and Methods

### 2.1. Study Design

Between January 2015 and December 2018, UGI endoscopy was performed to evaluate the severity of pooling secretions in the pharyngolaryngeal region in consecutive patients needing tube feeding for more than 4 weeks. Follow-up data were collected regarding subsequent pneumonia that required hospitalization. Patients were excluded due to age less than 20 years, pregnancy, or poor pharyngolaryngeal observations during endoscopy. The present study was approved by the Institutional Review Board of Tri-Service General Hospital, Taiwan. The patients were informed of the details of this study and allowed to participate after providing written informed consent.

### 2.2. Demographic and Clinical Data

Basic and clinical patient characteristics were recorded, such as age, gender, body mass index, serum albumin levels, reasons for tube feeding, duration of tube feeding, swallowing level scale score, and pneumonia requiring admission. The swallowing level scale score was assessed using the American Speech-Language-Hearing Association National Outcome Measurement System [21]. The swallowing level scale scores range between 1–7, with lower numbers indicating greater oral intake limitation and increased risk of pneumonia. Specifically, swallowing level scale scores between 1 and 3 are typically seen in tube-dependent patients, and those between 4 and 7 are usually seen in those on total oral intake. Pneumonia was diagnosed based on radiological evidence of pulmonary consolidation, serum white cell count >10,000/mm^3^, temperature >38 °C, shortness of breath, and required hospital admission [22].

### 2.3. UGI Endoscopy with Pharyngolaryngeal Observations

The choice of premedication was dependent on the preference of the endoscopist. Most of the endoscopic procedures were performed using topical anesthesia without intravenous sedation. Patients fasted for at least 4 h and were placed in the left lateral decubitus position. The tip of the UGI endoscope was inserted through a mouthpiece with its axis aligned with that of the patient’s esophagus. With advancement of the endoscope along the midline of the palate, the uvula could be visualized over the base of the tongue. The scope was rotated a little, past the uvula, and gently advanced with anterior flexion to visualize the pyriform sinus, laryngeal vestibule, vocal cords, and upper part of the trachea (Figure 1). A digital video recorder (HVO-550MD; Sony, Tokyo, Japan) was connected to the monitoring system of the endoscope.

Patients were evaluated according to the endoscopic observation of the amount and location of accumulated pooling secretions in the pharyngolaryngeal region (Figure 2) [23,24,25]. The severity of pooling secretions in the pharyngolaryngeal region was stratified into 3 groups: control group (<25% of pooling secretions filling the pyriform sinus), pharyngeal group (25–100% of pooling secretions filling the pyriform sinus but not entering the laryngeal vestibule), and laryngeal group (pooling secretions entering the laryngeal vestibule) [17].

### 2.4. Statistical Analysis

Statistical analyses were performed using SPSS 22.0 (IBM Inc., Armonk, NY, USA). Parametric continuous data were compared by analysis of variance (ANOVA). Categorical data were compared using the *χ*^2^ test and Yate’s correction or Fisher’s exact test. Regression analyses were performed to calculate the adjusted odds ratios with 95% confidence intervals (CI) for the risk associated with pneumonia. Multivariate regression analyses were used to assess the risk of pneumonia with adjustment for the age, gender, body mass index, serum albumin levels, reasons for tube feeding, duration of tube feeding, and swallowing level scale score as potential confounding factors. *P* value < 0.05 was considered statistically significant.

## 3. Results

### 3.1. Demographic and Clinical Data

Patients with PEG (*n* = 130) and NGT (*n* = 97) for long-term enteral feeding were enrolled (Table 1). There were no significant differences between the baseline characteristics of the PEG and NGT groups, regarding age, gender, body mass index, serum albumin level, duration of tube feeding, swallowing level scale score, reasons for tube feeding, severity of pooling secretions, occurrence of pneumonia, and incidence of pneumonia requiring hospital admission.

Statistical analysis associations of NGT or PEG according to the severity of pooling secretions are given in Table 2. There were no statistically significant associations between the presence of NGT or PEG with the severity of pooling secretions in the pharyngolaryngeal region.

### 3.2. Incidence of Pneumonia

The incidence of pneumonia requiring hospital admission in patients with NGT and PEG long-term tube feeding was analyzed by subgroup according to severity of pooling secretions (Table 3). The incidence of pneumonia (episodes/person-years) was significantly higher in patients with NGT versus PEG for the patients in the pharyngeal group (3.6 ± 1.0 vs. 2.3 ± 2.1, *P* < 0.001) and the laryngeal group (3.8 ± 0.5 vs. 2.3 ± 2.2 vs, *P* < 0.001), but not for patients in the control group (2.1 ± 0.5 vs. 1.8 ± 1.7, *P* = 0.362).

### 3.3. Risk Factor of Pneumonia

Multivariable regression analysis (Table 4) demonstrated that the risk of pneumonia requiring hospital admission was significantly higher in patients with NGT compared to patients with PEG (adjusted odds ratio = 2.85, 95% CI: 1.46–4.98, *P* < 0.001). The risk of pneumonia was significantly lower in patients with esophageal disorders (adjusted odds ratio = 0.35, 95% CI: 0.17–0.77, *P* = 0.010), but significantly higher in patients with neurological disorders (adjusted odds ratio = 1.27, 95% CI: 1.09–2.31, *P* = 0.042) and head and neck tumors (adjusted odds ratio = 1.34, 95% CI: 1.10–2.45, *P* = 0.038).

The risk of pneumonia was significantly increased in both the pharyngeal group (adjusted odds ratio = 1.25, 95% CI: 1.03–2.10, *P* = 0.048) and the laryngeal group (adjusted odds ratio = 1.79, 95% CI: 1.18–2.30, *P* = 0.017), compared to the control group.

### 3.4. Cumulative Proportion of Pneumonia

From the Kaplan–Meier estimate, the cumulative proportion of pneumonia (%) requiring hospital admission in patients with NGT was slightly higher than those with PEG, but did not reach statistical significance, *P* = 0.671. The cumulative proportion of pneumonia in all three groups (laryngeal, pharyngeal, control) in patients with NGT was slightly higher than those with PEG, but it did not reach statistical significance (*P* = 0.067, *P* = 0.345, and *P* = 0.792, respectively) (Figure 3).

The cumulative proportion of pneumonia requiring hospital admission was significantly higher in patients with NGT than in those with PEG for patients combining the pharyngeal and laryngeal groups (*P* = 0.035), but not for patients in the control group, *P* = 0.792 (Figure 4).

## 4. Discussion

To our knowledge, this is the first study to demonstrate: (a) a significantly higher incidence of pneumonia in patients with NGT than in PEG for abnormal amounts of pooling secretions accumulation in the pyriform sinus (pharyngeal group) or leak into the laryngeal vestibule (laryngeal groups), (b) a significantly higher risk factor for the pneumonia in patients with NGT than in PEG, and (c) a significantly higher cumulative proportion of pneumonia in patients with NGT than in PEG for patients in the pharyngeal and laryngeal groups compared to control.

Traditional diagnosis of oropharyngeal dysphagia is reached by clinical feeding evaluation, along with either fiberoptic endoscopic evaluation of swallow (FEES) or videofluoroscopic evaluation of swallow [26]. Symptoms of oropharyngeal dysphagia include coughing, choking, drooling, and regurgitation when swallowing liquids or solid food [19]. Absence of cough and airway protective responses renders oropharyngeal aspiration difficult to detect [27]. Lack of access to these tools often results in underrating or even neglecting the issue of dysphagia [28]. There is a need for a practical tool, such as UGI endoscopy with pharyngolaryngeal observations, for every patient with long-term tube feeding who is at risk of aspiration pneumonia. UGI endoscopy with pharyngolaryngeal observations is modified from FEES and has three advantages over FEES for the patients on long-term tube feeding.

(A) Patients do not need to have sufficient cognitive and physical skills: FEES can be carried out only when the patient has sufficient cognitive and physical skills to undergo testing with a modified texture of food or liquid [20,29]. UGI endoscopic pharyngolaryngeal observations can be routinely used for patients who do not have sufficient cognitive abilities, especially for many tube-feeding patients with concomitant advanced neurological diseases such as stroke, dementia, and Parkinson’s disease.

(B) Left lateral decubitus vs. upright position: Patients were placed in the left lateral decubitus position during the UGI endoscopy with pharyngolaryngeal observations, which differs from the upright position with an adjustable chair used in FEES [25,30]. UGI endoscopy provides clinical information closer to the patient’s real-world situations, especially for those patients lying in bed or with neurological conditions. When patients are placed in the left lateral decubitus position, endoscopic observation of the accumulated pooling secretions might fill the lowermost area of the right side of the pyriform sinus, leak into the laryngeal vestibule or vocal cords, and flow into the left side of the pyriform sinus (Figure 2C,D).

(C) Fasting vs. test foods challenge: Patients were fasted before and during the UGI endoscopy with pharyngolaryngeal observations. FEES requires a feeding challenge with test foods of different consistencies. Daily production of 1000–1500 mL of saliva may transiently present in the oral cavity [18]. For patients with normal swallowing function, pooling secretions can be minimized by swallowing multiple times. UGI endoscopy is widely used for diagnosis or treatment for dysphagic patients who need long-term tube feeding. UGI endoscopic pharyngolaryngeal observations allow for direct observation of abnormal pooling secretions in the pharyngolaryngeal region, and indicate which patients may be at the risk for the subsequent development of aspiration pneumonia during long-term tube feeding.

A Cochrane systematic review and meta-analysis of the impact of PEG versus NGT feeding for 645 adults with swallowing disturbance did not favor PEG for improved pneumonia outcomes due to high levels of statistical heterogeneity (*I*^2^ = 81%) [2]. To address heterogeneity, subgroup analyses was performed in this study. PEG versus NGT were well-matched in terms of the age, gender, body mass index, serum albumin level, swallowing level scale score, and reasons for tube feeding. Comparing PEG and NGT, we found that the incidence of pneumonia and cumulative proportion of pneumonia were significantly higher in patients with oropharyngeal dysphagia, but not in patients without oropharyngeal dysphagia (control group). This result suggests that risk of pneumonia favors PEG over NGT in patients with oropharyngeal dysphagia.

The development of aspiration pneumonia depends on the cough reflex, volume and pH level of aspirated material, and the integrity of the immune system [27]. NGT that passes through the gastroesophageal sphincter may increase gastroesophageal reflux [31] and NGT that passes through the upper esophageal sphincter may interfere with the protective cough reflexes, thereby increasing the risk of aspiration [32,33,34].

Limitations: First, randomized controlled trials are ideal when comparing interventions, but given ethical and logistical challenges, randomized trials on enteral feeding are unlikely to be performed in current clinical practice [35]. Aspiration pneumonia occurs when food, saliva, pharyngeal secretions, or gastric contents are breathed into the lungs [8,36,37,38]. Therefore, the degree that the aspiration of colonized oropharyngeal contents contributes to pneumonia is uncertain [39,40]. It was not possible to identify the cause of pneumonia that required admission; rather, it was the reflux of gastric contents or oropharyngeal pooling secretions. Finally, this study was performed at a single tertiary care hospital, so generalizability may be limited due to small sample size. One of this study’s strengths is that the long follow-up period may help reduce this potential effect.

## 5. Conclusions

PEG is a better choice for long-term tube feeding compared to NGT due to the decreased risk of developing pneumonia requiring hospital admission, especially for patients with abnormal amounts of secretions accumulation in the pyriform sinus or leak into the laryngeal vestibule. These findings are useful in the development of clinical guidelines and to inform discussions among health care providers, patients, and family members with regards to long-term feeding methods.

## Figures and Tables

**Figure 1 nutrients-11-02969-f001:**
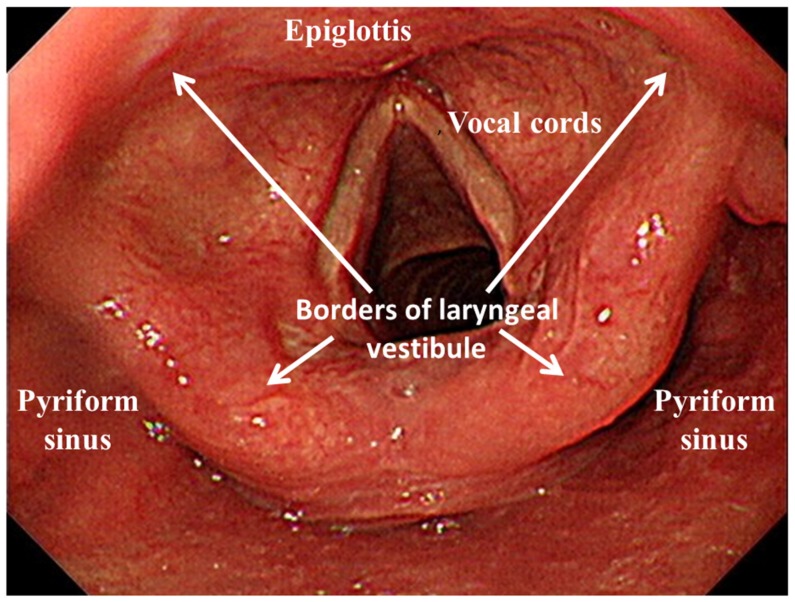
Endoscopic view of the pharyngolaryngeal region.

**Figure 2 nutrients-11-02969-f002:**
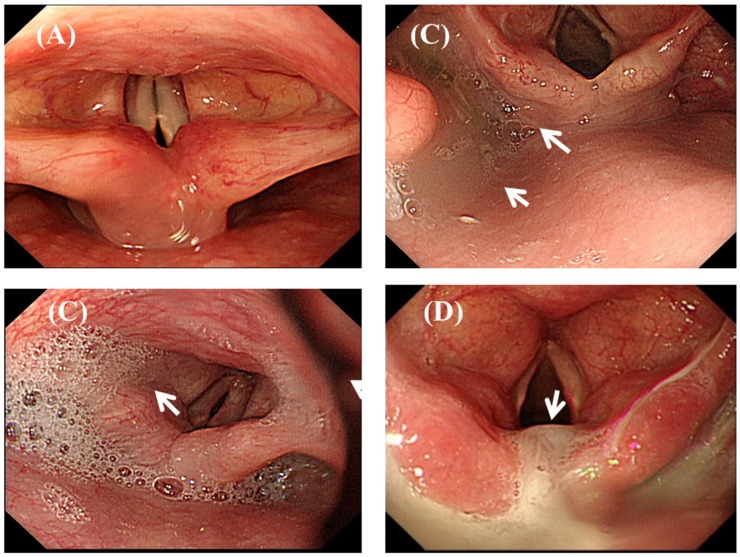
Endoscopic views of pooling secretions (arrow) in the pharyngolaryngeal region. Absence of pooling secretions filling the right side pyriform sinus (**A**), pooling secretions filling the right side pyriform sinus (**B**), and thin pooling secretions (**C**) or thick secretions (**D**) entering the laryngeal vestibule.

**Figure 3 nutrients-11-02969-f003:**
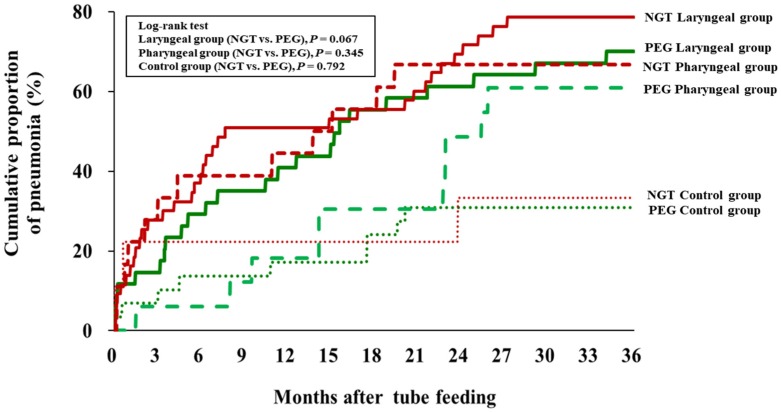
Cumulative proportion of pneumonia of 3 groups (control, pharyngeal, and laryngeal groups) of percutaneous endoscopic gastrostomy (PEG) versus nasogastric tube (NGT).

**Figure 4 nutrients-11-02969-f004:**
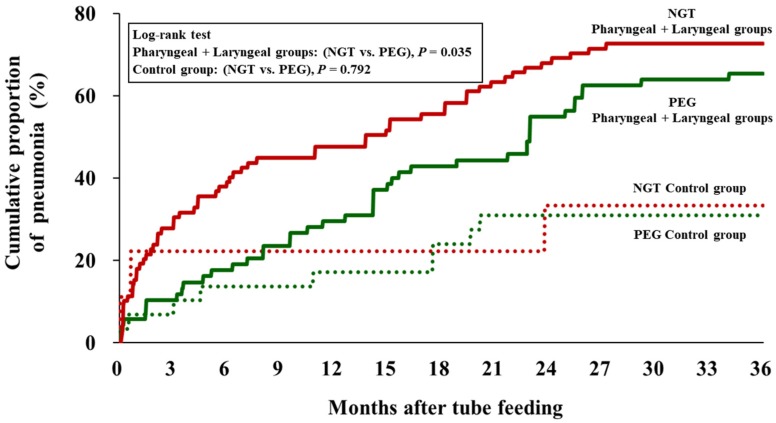
Cumulative proportion of percutaneous endoscopic gastrostomy (PEG) versus nasogastric tube (NGT) feeding with (pharyngeal + laryngeal groups) and without (control) visual evidence of oropharyngeal dysphagia.

**Table 1 nutrients-11-02969-t001:** Patients characteristics.

Variable	NGT (*n* = 130)	PEG (*n* = 97)	*P* Value
Age (years)	67.1 ± 15.2	70.4 ± 14.7	0.102
Gender			0.268
Female	51 (39.2%)	31 (32.0%)	0.511
Male	79 (60.8%)	66 (68.0%)	0.368
Body mass index (kg/m^2^)	21.9 ± 4.1	22.3 ± 4.0	0.501
Albumin (gm/dl)	3.1 ± 0.5	3.2 ± 0.5	0.925
Swallowing level scale score	2.0 ± 0.9	2.1 ± 0.9	0.447
Reasons for tube feeding, no (%)			0.153
Esophageal disorders	48 (36.9%)	38 (39.2%)	0.872
Neurological disorders	50 (38.5%)	45 (46.4%)	0.436
Head and neck tumors	32 (24.6%)	14 (14.4%)	0.438
Duration (days)	741 ± 361	785 ± 338	0.348
Pooling secretions severity, no (%)			0.275
Control group	81 (62.3%)	58 (59.8%)	0.765
Pharyngeal group	27 (20.8%)	15 (15.5%)	0.674
Laryngeal group	22 (16.9%)	24 (24.7%)	0.516
Occurrence of pneumonia, no (%)	46 (35.4%)	31 (32.0%)	0.671
Incidence of pneumonia (episodes/person-years)	2.3 ± 3.4	2.0 ± 3.1	0.225

NGT: Nasogastric tube, PEG: Percutaneous endoscopic gastrostomy, Swallowing level scale score: American Speech-Language-Hearing Association National Outcome Measurement System.

**Table 2 nutrients-11-02969-t002:** Association of the NGT or PEG feeding with the severity of pooling secretions.

Pooling Secretions Severity	NGT (*n* = 130)	PEG (*n* = 97)	Crude Odds Ratio (95% CI)	*P* Value	Adjusted Odds Ratio (95% CI)	*P* Value
Control group	81 (62.3%)	58 (59.8%)	Reference		Reference	
Pharynx group	27 (20.8%)	15 (15.5%)	1.35 (0.68–2.71)	0.395	1.04 (0.39–2.78)	0.935
Larynx group	22 (16.9%)	24 (24.7%)	1.73 (0.87–3.43)	0.116	1.59 (0.68–3.75)	0.285

NGT: Nasogastric tube, PEG: Percutaneous endoscopic gastrostomy, adjusted for variables listed in Table 1.

**Table 3 nutrients-11-02969-t003:** Incidence of pneumonia in patients with tube feeding.

Variable	NGT (*n* = 130)	PEG (*n* = 97)	*P* Value
Incidence of pneumonia (episodes/person-years)			
Control group	2.1 ± 0.5	1.8 ± 1.7	0.362
Pharyngeal group	3.6 ± 1.0	2.3 ± 2.1	<0.001
Laryngeal group	3.8 ± 0.5	2.3 ± 2.2	<0.001

NGT: Nasogastric tube, PEG: Percutaneous endoscopic gastrostomy.

**Table 4 nutrients-11-02969-t004:** Multivariable analysis of the risk factors of pneumonia.

Variable	With Pneumonia (*n* = 77)	Without Pneumonia (*n* = 150)	Crude Odds Ratio (95% CI)	*P* Value	Adjusted Odds Ratio (95% CI)	*P* Value
Age (years)	70.6 ± 14.3	67.4 ± 15.4	1.05 (0.92–1.09)	0.126	1.03 (0.98–1.05)	0.129
Gender						
Female	25 (32.5%)	57 (38.0%)	Reference		Reference	
Male	52 (67.5%)	93 (62.0%)	1.28 (0.71–2.28)	0.467	1.21 (0.55–2.14)	0.426
NGT vs. PEG						
PEG	31 (40.3%)	66 (44.0%)	Reference		Reference	
NGT	46 (59.7%)	84 (56.0%)	2.88 (1.63–5.08)	<0.001	2.85 (1.46–4.98)	<0.001
Body mass index (kg/m^2^)	21.6 ± 4.2	22.3 ± 4.0	0.97 (0.91–1.04)	0.223	0.99 (0.91–1.08)	0.835
Albumin (gm/dL)	3.1 ± 0.5	3.4 ± 0.6	0.36 (0.21–0.65)	0.002	0.49 (0.21–1.43)	0.168
Swallowing level scale score	2.0 ± 1.3	3.1 ± 2.3	0.72 (0.60–0.88)	0.001	0.91 (0.70–1.18)	0.433
Reasons for feeding tube, no (%)						
Esophageal disorders	17 (22.1%)	69 (46.0%)	0.48 (0.28–0.85)	0.004	0.35 (0.17–0.77)	0.010
Neurological disorders	39 (50.6%)	56 (37.3%)	1.36 (1.11–2.78)	0.001	1.27 (1.09–2.31)	0.042
Head and neck tumors	21 (27.3%)	25 (16.7%)	1.63 (1.20–2.94)	0.001	1.34 (1.10–2.45)	0.038
Pooling secretions severity, no (%)						
Control group	32 (41.6%)	107 (71.3%)	Reference		Reference	
Pharynx group	23 (29.8%)	23 (15.3%)	3.34 (1.57–6.66)	0.001	1.24 (0.99–2.08)	0.051
Larynx group	22 (28.6%)	20 (13.4%)	3.66 (1.63–7.99)	0.001	1.73 (1.16–2.24)	0.023

NGT: Nasogastric tube, PEG: Percutaneous endoscopic gastrostomy, Swallowing level scale score: American Speech-Language-Hearing Association National Outcome Measurement System, CI: Confidence interval.

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
