# Peer review of "Percutaneous Endoscopic Gastrostomy versus Nasogastric Tube Feeding: Oropharyngeal Dysphagia Increases Risk for Pneumonia Requiring Hospital Admission"

_nutrients, 2019, doi:10.3390/nu11122969_

Round 1
Reviewer 1 Report
This manuscript “Percutaneous Endoscopic Gastrostomy versus Nasogastric Tube Feeding: Oropharyngeal Dysphagia Increases Risk for Pneumonia Requiring Hospital Admission” aims to describe the outcomes of patients receiving an NG tube versus PEG with regard to incidence of pneumonia and oropharyngeal dysphagia. This is a very important topic. Additional clarifications would be helpful in enhancing the value of the manuscript:
Major revisions:
Materials and Methods: The method that the authors describe for diagnosing oropharyngeal dysphagia is not a validated measure. The authors can report on the three groups that they observed (i.e. percent of pooling secretions, etc) but until this methodology is compared with the gold standard (videofluoroscopic swallow study) it cannot be utilized to diagnose oropharyngeal dysphagia. Would suggest either restating your comparisons, removing the verbiage “oropharyngeal dysphagia” and replacing it with the descriptors of what was visualized on endoscopy or removing this component from the study and reporting on NG vs PEG and its correlation with pneumonia and hospitalizations alone. Results/Discussion: Suggest running statistical analysis (percentages are present without p-values) to determine if the two groups (NG vs PEG) are similar in terms of reason for tube feeding. Currently, it looks like there is a greater percentage of patients with head and neck tumors in the NGT group. Is this statistically significant? If so, could this account for the increase in number of patients with pneumonia in the NGT group? Results: Please show statistical significance of association between presence of NGT or PEG and association with pooling of secretions. Discussion: “Our study demonstrated that routine UGI can alert physicians of the presence of dysphagia… “ This study can alert physicians to the pooling of secretions but has not been validated as an assessment for dysphagia.Minor revisions:
Materials and Methods: Is the patient under sedation during the endoscopy? Discussion: Traditional diagnosis of oropharyngeal dysphagia is by clinical feeding evaluation along with either videofluoroscopic evaluation of swallow or fiberoptic endoscopic evaluation of swallow.
Author Response
Major revisions:
Materials and Methods: The method that the authors describe for diagnosing oropharyngeal dysphagia is not a validated measure. The authors can report on the three groups that they observed (i.e. percent of pooling secretions, etc) but until this methodology is compared with the gold standard (videofluoroscopic swallow study) it cannot be utilized to diagnose oropharyngeal dysphagia.
Answer: We appreciate the reviewer’s comment. We have revised the phrase “oropharyngeal dysphagia” in the Abstract, Introduction, Materials and Methods, Results, and Discussion sections, as needed. We reported on the three groups using the descriptors of what was visualized on endoscopy in the manuscript.
Would suggest either restating your comparisons, removing the verbiage “oropharyngeal dysphagia” and replacing it with the descriptors of what was visualized on endoscopy or removing this component from the study and reporting on NG vs PEG and its correlation with pneumonia and hospitalizations
Answer: Thank you for your comments. We agree and have removed the phrase “oropharyngeal dysphagia” and replace it with the descriptors of what was visualized on endoscopy in discussing our comparisons.
Results/Discussion: Suggest running statistical analysis (percentages are present without p-values) to determine if the two groups (NG vs PEG) are similar in terms of reason for tube feeding. Currently, it looks like there is a greater percentage of patients with head and neck tumors in the NGT group. Is this statistically significant? If so, could this account for the increase in number of patients with pneumonia in the NGT group?
Answer: We have run and have presented the statistical analysis. There were no significant differences between the baseline characteristics of the PEG and NGT groups in regards to reasons for tube feeding (esophageal disorders, P = 0.872; neurological disorders, P = 0.436; head & neck tumors, P = 0.438). (Table 1)
Results: Please show statistical significance of association between presence of NGT or PEG and association with pooling of secretions.
Answer: We now present the statistical analyses on associations between NGT and PEG with the severity of pooling secretions in Table 2. There were no statistically significant associations between presence of NGT or PEG with the severity of pooling of secretions in the pharyngolaryngeal region. (Page 4, line 31-33)
Discussion: “Our study demonstrated that routine UGI can alert physicians of the presence of dysphagia… “ This study can alert physicians to the pooling of secretions but has not been validated as an assessment for dysphagia.
Answer: We clarified this question in the Discussion section. UGI endoscopic pharyngolaryngeal observations allow for direct observation of abnormal pooling secretions in the pharyngolaryngeal region, and indicate which patients may be at the risk for the subsequent development of aspiration pneumonia during long-term tube feeding. (Page 8, line 33-36)
Minor revisions:
Materials and Methods: Is the patient under sedation during the endoscopy?
Answer: We clarified this question in the Materials and Methods. The choice of premedication was dependent on the preference of the endoscopist. Most of the endoscopic procedures were performed using topical anesthesia without intravenous sedation. (Page 2, line 33-34)
Discussion: Traditional diagnosis of oropharyngeal dysphagia is by clinical feeding evaluation along with either videofluoroscopic evaluation of swallow or fiberoptic endoscopic evaluation of swallow.
Answer: We appreciate the reviewer’s comment and took this opportunity to make an improvement of the manuscript.
We clarified the description of traditional diagnosis of oropharyngeal dysphagia for patients with long-term tube feeding in the Discussion section. (Page 8, line 3-11) We explained the UGI endoscopy with pharyngolaryngeal observations has three advantages which over FEES for patients with long-term tube feeding. Patients do not need to have sufficient cognitive and physical skills. (Page 8, line 12-17) Patients do not need to sit in upright position with an adjustable chair. (Page 8, line 18-25) Patients do not need the test foods challenge. (Page 8, line 26-34)

Reviewer 2 Report
Thank you for this important contribution to the literature on this topic. I have few improvements to offer to your manuscript.
Under limitations (p 7 out of 9) the sentence beginning with "Since" doesnt read clearly - I suggest you remove "Since" and start at "Aspiration."
Author Response
Reviewer #2:
Thank you for this important contribution to the literature on this topic. I have few improvements to offer to your manuscript. Under limitations (p 7 out of 9) the sentence beginning with "Since" doesnt read clearly - I suggest you remove "Since" and start at "Aspiration."
Answer: We appreciate the reviewer’s comment. We deleted the sentence beginning with "Since" and started at "Aspiration" in the limitation section. (Page 8, line 51)

Reviewer 3 Report
Review:
The authors of the manuscript under review, “Percutaneous endoscopic gastrostomy versus nasogastric tube feeding: oropharyngeal dysphagia increases risk for pneumonia requiring hospital admission”, compare percutaneous endoscopic gastrostomy (PEG) and nasogastric tube (NGT) feeding retrospectively in a cohort of 227 patients. The two groups were fairly comparable at the baseline and are further stratified following endoscopic examination. The authors observe that pneumonia is more frequent in NGT group than PEG group; and use multivariable analysis to show that PEG users have a lower risk of pneumonia requiring hospital admission than NGT users even when accounting for other factors. Finally, cumulative proportion of pneumonia may be elevated in NGT group.
The presented data offers valuable insight to dysphagia. The methods are sound and the statistical analyses suitable. The reasoning and presentation can be easily followed. The manuscript has two main shortcomings: 1) the cumulative proportion of pneumonia is computed by combining two of the three stratified groups without justification and hence the abstract seems to overstate the actual results; 2) the discussion is weak compared to the rest of the manuscript and would require further editing.
The manuscript presents interesting results that are relevant to the scientific community. The overall quality of the research performed is good. In its current form, the manuscript is not suitable for publication. However, the necessary changes are minor. A suggested list of improvements is provided as an appendix to this letter.

Author Response
The cumulative proportion of pneumonia is computed by combining two of the three stratified groups without justification and hence the abstract seems to overstate the actual resultsAnswer: We appreciate the reviewer’s comment. We corrected this in the Abstract and Discussion sections.
PEG is a better choice than NGT feeding due to the decrease in risk of pneumonia requiring hospital admission, especially for patients with abnormal amounts of secretions accumulation in the pyriform sinus (pharyngeal group) or leak into the laryngeal vestibule (laryngeal groups). (Page 1, line 27-30; Page 8, line 1-2; Page 9, line 7-9)
The discussion is weak compared to the rest of the manuscript and would require further editing.
Answer: We appreciate the reviewer’s comment. Accordingly, we have made the following changes in the Discussion section.
We clarified the description of traditional diagnosis of oropharyngeal dysphagia in the Discussion section. (Page 8, line 3-11) We explained the UGI endoscopy with pharyngolaryngeal observations has three advantages over FEES for patients with long-term tube feeding. Patients do not need to have sufficient cognitive and physical skills. (Page 8, line 12-17) Patients do not need to sit in upright position with an adjustable chair. (Page 8, line 18-25) Patients do not need the test foods challenge. (Page 8, line 26-34) We corrected the overstatement of our results of the cumulative proportion of pneumonia. (Page 8, line 1-2; Page 9, line 7-9)The manuscript presents interesting results that are relevant to the scientific community. The overall quality of the research performed is good. In its current form, the manuscript is not suitable for publication. However, the necessary changes are minor. A suggested list of improvements is provided as an appendix to this letter.
Answer: We have addressed the suggested list of improvements
Abstract
Removed the phrase “oropharyngeal dysphagia” and replaced it with descriptors of what was visualized on endoscopy. (Page 1, line 16) Corrected the overstatement of the results of the cumulative proportion of pneumonia. (Page 1, line 27-30)Introduction
Removed the phrase “oropharyngeal dysphagia” as appropriate and replaced it with descriptors of what was visualized on endoscopy. (Page 2, line 10-11)Materials and Methods
Removed the phrase “oropharyngeal dysphagia” and replaced it with descriptors of what was visualized on endoscopy. (Page 2, line 15; Page 2, line 42-44) Clarified the question whether patient under sedation during the endoscopy. (Page 2, line 33-34)Results
Removed the phrase “oropharyngeal dysphagia” and replaced it with the descriptors of what was visualized on endoscopy. (Page 4, line 15; Table 1; Table 4) Present statistical analysis to determine whether the two groups (NG vs PEG) are similar in terms of reason for tube feeding. (Table 1) Present statistical analysis of the association between NGT or PEG with the severity of pooling secretions in the pharyngolaryngeal region. (Page 4, line 31-33; Table 2)Discussion
Removed the phrase “oropharyngeal dysphagia” and replaced it with descriptors of what was visualized on endoscopy. (Page 7, line 13-15; Page 8, line 1-2; Page 9, line 8-9) Clarified the description of diagnosis of oropharyngeal dysphagia. (Page 8, line 3-11) Explained that UGI endoscopy with pharyngolaryngeal observations has three advantages over FEES for patients with long-term tube feeding. Patients do not need to have sufficient cognitive and physical skills. (Page 8, line 12-17) Patients do not need to sit in upright position with an adjustable chair. (Page 8, line 18-25) Patients do not need test foods challenge. (Page 8, line 26-34) Deleted the sentence beginning with "Since" and started at "Aspiration" in the limitation section. (Page 8, line 51) Corrected the overstatement of the results of the cumulative proportion of pneumonia. (Page 8, line 1-2; Page 9, line 7-9)
Round 2
Reviewer 1 Report
I accept the revised manuscript for publication in its current form.